**Data Availability Statement:** The cost data used in this analysis were obtained from the Korean national health insurance's (KNHIS) database and approved for academic use. Legal restrictions exist

# A tool to measure the impact of inaction toward elimination of hepatitis C: A case study in Korea

Yong Kyun Won[1], Kyung Sik Kang[1], Yuri Sanchez Gonzalez[2], Homie Razavi[3], Ellen Dugan[3], Kwang-Hyub Han[4], Sang Hoon Ahn[4], Mi Young Jeon[4], Do Young Kim[4]*

**1** AbbVie Korea, Ltd., Seoul, Korea, **2** AbbVie Inc., Mettawa, Illinois, United States of America, **3** Center for Disease Analysis Foundation (CDAF), Lafayette, Colorado, United States of America, **4** Department of Internal Medicine, Yonsei University, College of Medicine, Seoul, Korea

* dyk1025@yuhs.ac

## Abstract

### Background and aims

Hepatitis C virus (HCV) and its sequelae present a significant source of economic and societal burden. Introduction of highly effective curative therapies has made HCV elimination attainable. The study used a predictive model to assess the clinical and economic impact of implementing national screening and treatment policies toward HCV elimination in Korea.

### Methods

A previously validated Markov disease progression model of HCV infection was employed to analyze the clinical and economic impact of various strategies for HCV diagnosis and treatment in Korea. In this analysis, the model compared the clinical and economic outcomes of current HCV-related interventions in Korea (7,000 patients treated and 4,200 patients newly diagnosed annually, starting in 2017) to four elimination scenarios: 1) initiating sufficient diagnosis and treatment interventions to meet the World Health Organization's GHSS elimination targets by 2030, 2) delaying initiation of interventions by one year, 3) delaying initiation of interventions by two years and 4) accelerating initiation of interventions to meet elimination targets by 2025. Modelled historical incidence of HCV was calibrated to match a viremic HCV prevalence of 0.44% in 2009. Elimination scenarios required 24,000 treatments and 34,000 newly diagnosed patients annually, starting in 2018, to reach the 2030 targets.

### Results

Compared to current "status quo" interventions, elimination (or accelerated elimination by 2025) would avert 23,700 (27,000) incident cases of HCV, 1,300 (1,400) liver-related deaths (LRDs) and 2,900 (3,100) cases of end-stage liver disease (ESLD) over the 2017–2030 time period. Postponing interventions by one (or two) years would avert 21,100 (18,600) new HCV infections, 920 (660) LRDs and 2,000 (1,400) cases of ESLD by 2030. Following elimination or accelerated elimination strategies would save 860 million USD or 1.1 billion

on the sharing of KNHIS because the dataset holds sensitive information and data may be identifiable at the regional or personal level. Therefore, the author restricts the right for data access. DY Kim, corresponding author, has obtained approval for this research from his center's institutional review board (https://ocr.yuhs.ac/HPC/HPCIndex.aspx, Tel:+82-2-2228-0430, E-mail: irb@yuhs.ac). Future researchers can request access to the data underlying your study from DY Kim, Corresponding author (DYK1025@yuhs.ac). Data from KNHIS was also obtained under this IRB approval. The authors will also attach the SAS code used in the extraction of KNHIS data. The data file analyzed using the code is not able to be opened to the public by KNHIS's regulation. The point of contact where the data requests was done and can be made further in KNHIS is as follow (Mr. HC Yoon, Data operation team #3, Dept. of Big data, Health insurance policy institute, KNHIS., Tel: +82-33-736-2473, (+82-33-811-2000 (English call center for foreigners)) Fax: +82-33-749-6337 E-mail: sapsary@nhis.or.kr).

**Funding:** The design, study conduct, analysis, and financial support for the study were provided by AbbVie. AbbVie participated in the interpretation of data, review, and approval of the publication. All authors had access to all relevant data.

**Competing interests:** I have read the journal's policy and the authors of this manuscript have the following competing interests. Homie Razavi is an employee of Center for Disease Analysis (CDA). CDA has received funding from AbbVie Inc. for this project. CDA has also received research funding from AbbVie, Gilead, Intercept, and Pfizer. Ellen Dugan, Kwang-Hyub Han, Sang Hoon Ahn, Mi Young Jeon and Do Young Kim have nothing to declare. Yuri Sanchez Gonzalez, Kyung Sik Kang, Yong Kyun Won are employees of AbbVie and may own stock/stock options.

USD by 2030, respectively, compared to the status quo, requiring an up-front investment in prevention that decreases spending on liver-related complications and death.

## Conclusions

By projecting the impact of interventions and tracking progress toward GHSS elimination targets using modelling, we demonstrate that Korea can prevent significant morbidity, mortality and spending on HCV. Results should serve as the backbone for policy and decision-making, demonstrating how aggressive prevention measures are designed to reduce future costs and increase the health of the public.

## Introduction

Chronic hepatitis C (CHC) is a leading cause of liver disease affecting 71 million people in 2017, representing 1% of the global population [1]. Hepatitis C virus (HCV) has received increased attention in South Korea (hereafter Korea) in recent years due to a notable number of healthcare-associated outbreaks reported between 2015 and 2016, associated with unsafe injection practices [2]. Due to factors including delayed viral identification, absence of a vaccine, costly treatment and the lack of a national database for HCV in Korea, CHC infection has proliferated in the Korean population, and only recent advancements have been able to estimate the true disease burden. In 2009, the anti-HCV prevalence in the adult Korean population was 0.78% (approximately 385,000 anti-HCV cases) [3].

The introduction of direct-acting antiviral agents (DAAs) with a ≥90% cure rate in 2014 caused a dramatic shift in CHC awareness and treatment and has made HCV elimination possible. As such, the World Health Organization introduced Global Health Sector Strategy (GHSS) targets for the elimination of HCV as a public health threat by 2030, including 1) a 90% reduction in incident cases of CHC, 2) a 65% reduction in HCV-related deaths, 3) diagnosis of 90% of chronic infections and 4) treatment of 80% of eligible people with CHC [4]. Prior to the launch of DAA regimens (which were approved by the national healthcare system in Korea in 2015), only 14% of Korean CHC patients were estimated to have received treatment with an interferon-based antiviral regimen over a 5-year period [5]. Today, pan-genotypic regimens are reimbursable without restrictions (i.e. patients of all fibrosis stages are eligible for treatment).

Due to the largely asymptomatic nature of HCV, infections are often diagnosed after the manifestation of late-stage complications. Late-stage disease can be exponentially more expensive to manage and treat, which is an incentive to screen and diagnose patients in earlier stages of disease. Currently, there is no systematic screening system to find asymptomatic CHC patients in Korea. Up to 75% of HCV-infected Koreans are unaware of their infection and potential to transmit; therefore, screening and treatment are integral components of a plan toward elimination [6]. A predictive model was used to inform national screening and treatment policies to achieve HCV elimination in Korea. The primary objectives were to 1) report the current state of HCV management in Korea (status quo) and 2) identify the clinical and economic efforts required for Korea to be on track to eliminate HCV by 2030.

## Methods

This analysis was based on a previously published Markov disease progression model that forecasts the annual HCV-infected population in a country or region by stage of liver disease, sex and age [7]. The Impact of Inaction (IOI) model, used in the present analysis, is a streamlined version of the disease progression model with the added features of population scalability and economic analysis. Korean demographic, epidemiological and economic data on HCV disease burden and spending were input into the model to quantify the disease burden and medical costs associated with CHC infection and its sequelae. Various intervention scenarios were modelled to assess the clinical and economic impact of delaying the diagnosis and treatment of HCV infection. This study was designed from the Korean formal healthcare sector perspective with a target population of all HCV-infected Koreans. Study parameters were analyzed over a 14-year time horizon of 2017–2030. All costs were discounted at the annual rate of 3%.

### Model structure

The Microsoft Excel® based disease burden model, developed by the Center for Disease Analysis (CDA), tracks the sex- and age-structured HCV-infected population from incidence through stages of liver disease (fibrosis (F0–F3 on the METAVIR scale), compensated cirrhosis, decompensated cirrhosis, hepatocellular carcinoma and liver transplantation) to eventual death (all-cause or HCV-liver-related). The validation of modelled outcomes has been previously published [7].

### HCV epidemiology and healthcare cost data

In 2009, there were about 49.4 million inhabitants in Korea, with an estimated 216,000 RNA positive cases of HCV (0.44% viremic prevalence). This was calculated by adjusting an anti-HCV prevalence of 0.78% by 56.1% viremia (Table 1) [3]. Modelled historical incidence of HCV was calibrated to match the reported prevalence. The diagnosis coverage was estimated at 30%, or 64,800 HCV-infected people [6]. Currently in Korea, HCV treatment is reimbursed for all fibrosis stages, and there is no age limit for treatment eligibility. From 2010 through 2015, 4,600 patients were treated annually [8]. The national estimate of a 90% sustained virologic response at 12 weeks or more post-treatment (SVR12) was assumed.

   The Korean healthcare system provides universal coverage for all inhabitants of Korea following at least six months of residency. The Korean National Health Insurance System (KNHIS) captures all healthcare visits and claims data and is therefore considered representative of the HCV-infected population seeking healthcare in Korea [14]. Cases of HCV-related death and end-stage liver disease (ESLD) indicators (decompensated cirrhosis (DC), liver transplants (LT) and hepatocellular carcinoma (HCC)) extracted from the KNHIS system were used to calculate average cost data.

   Average annual spending on laboratory testing, treatment and diagnosis of liver-related complications and extra-hepatic manifestations (EHM) were calculated from the 2017 KNHIS claims database. The cost of treating HCV extrahepatic manifestations used in this study is based on data from the KNHIS (Study Number: NHIS-2019-1-140). The 2017 cost data were extracted for patients with either a principal diagnosis-I or sub diagnosis-I of CHC or EHM. The ICD-10-CM Code used for CHC was B18.2. A maximum of three EHM diagnoses per patient were considered in the calculation of EHM-related costs. HCV treatment (including treatment assessment and monitoring) costs were estimated at 9,540 USD per patient, and the cost of HCV diagnostic testing was 12 USD (Table 1). Prices for all healthcare services in Korea are fixed by KNHIS and therefore uniform across all cases. Outpatient and inpatient service costs are equal as all HCV-related costs are paid for by public funds.

**Table 1. HCV-related inputs for Korean impact of inaction disease burden model.**

| Category | Item | Year | Value | Source |
|---|---|---|---|---|
| **Disease burden model parameters** | Anti-HCV prevalence | 2009 | 0.78% | [3] |
| | Viremic rate (HCV-RNA positive) | 2009 | 56.1% | [3] |
| | Viremic diagnosed | 2009 | 64,800 (30% diagnosis coverage) | [6] |
| | Genotype distribution | 2013 | 52.7% genotype 1 | [9] |
| | Treated | 2010–2017 | 7,000 in 2017 | [8] |
| | Liver transplants | 1992–2000, 2000–2012 | 1,200 in 2013 | [10, 11] |
| | Liver transplants due to HCV | 2009–2013 | 5% | [10] |
| **Diagnostic and laboratory costs per test (USD)** | Anti-HCV test | 2016 | 12 | [12] |
| | RNA/ PCR test | 2016 | 90 | [12] |
| | HCV genotyping | 2016 | 102 | [12] |
| | Staging/liver biopsy/fibroscan | 2016 | 64 | [12] |
| **Screening costs as part of treatment (USD)** | Cost per screen | 2016 | 12 | [5] |
| | Lab costs for F0-F3 | 2016 | 243 | [5] |
| | Lab costs for ≥F4 | 2016 | 243 | [5] |
| **Annual costs per diagnosed patient (USD)** | Follow-up costs for F0-F3 | 2016 | 888 | [5] |
| | Post-SVR12 healthcare costs for F0-F3 | 2016 | 23 | [5] |
| | Compensated cirrhosis | 2016 | 1,132 | [5] |
| | Decompensated cirrhosis | 2016 | 5,916 | [5] |
| | Hepatocellular carcinoma | 2016 | 6,831 | [5] |
| | Liver transplant | 2016 | 66,831 | [5] |
| | Post-liver transplant | 2016 | 6,592 | [5] |
| | Extra-hepatic manifestations* | 2017 | 693 | [13] |

*Extra-hepatic manifestations include: Arthritis, Depression, End-Stage Renal Disease, Glomeruloneprhitis, Heart Failure, Lichen Planus, Lymphoma, Myocardial Infarction, Mixed Cryoglobulinemia, Porphyria Cutanea Tarda, Sjögren Syndrome, Stroke, Type II Diabetes, Cardiovascular Disease, and Chronic Renal Disease

## Scenario development

Five scenarios were developed to estimate the impact of various interventions on the disease burden of CHC in Korea. The status quo served as a baseline to calculate cases of ESLD and HCV-related deaths averted by 2030 for each elimination scenario. For the status quo scenario, over 2016–2020, a 50% reduction in the number of treated patients was applied to simulate the expected depletion of diagnosed patients and patients under care in the first few years of DAA treatment, a trend demonstrated across Western countries. The status quo scenario did not meet GHSS elimination targets, therefore, four elimination scenarios were designed to meet these targets while also taking into consideration the impact of delayed intervention [4].

Elimination scenarios were modelled with changes to the start year of intervention, number of diagnosed cases and number of cases treated (Table 2). Unless specified below, elimination scenarios assumed a SVR12 of 90%, treatment eligibility for all ages and were designed to meet GHSS elimination targets by 2030. The five modelled scenarios are as follows:

1. **Status quo**: Modelled from the most recent available national data (starting in 2017), estimating 4,200 newly diagnosed patients and 7,000 patients treated annually.

2. **Standard elimination**: 24,000 newly diagnosed patients and 34,000 patients treated annually, beginning in 2018.

3. **One-year delayed elimination**: 24,000 newly diagnosed patients and 34,000 patients treated annually, beginning in 2019.

**Table 2. HCV-related intervention inputs implemented annually by scenario, 2017–2030[*].**

| Scenarios | Start year | Newly diagnosed patients | Annual treated patients |
|---|---|---|---|
| Status quo | 2017 | 4,200 | 7,000 |
| Elimination | 2018 | 24,000 | 34,000 |
| One-year delayed elimination | 2019 | 24,000 | 34,000 |
| Two-year delayed elimination | 2020 | 24,000 | 34,000 |
| Accelerated elimination | 2018 | 42,000 | 52,000 |

[*]Assume no fibrosis stage restrictions (≥F0), all ages eligible for treatment and an average sustained virologic response after 12 weeks (SVR12) of 90% for all scenarios.

4. **Two-year delayed elimination**: 24,000 newly diagnosed patients and 34,000 patients treated annually, beginning in 2020.

5. **Accelerated elimination**: 42,000 newly diagnosed patients and 52,000 patients treated annually, beginning in 2018. This scenario was designed to meet GHSS elimination targets by 2025.

## Results

### Overall clinical and economic outcomes

An estimated 169,000 Koreans were infected with viremic HCV in 2017, of which 40% (n = 70,000) had previously been diagnosed. About 4% (n = 6,300) of all infected Koreans received treatment and achieved cure in 2017. Following the status quo, about 63,000 infected Koreans are expected to remain undiagnosed and untreated by 2030. The four elimination scenarios are projected to result in the diagnosis of all infected Koreans by 2030, with 100 or fewer patients awaiting treatment. Compared to the status quo, elimination (regardless of delay) also resulted in a decreased disease burden and lesser overall spending over 2017–2030. However, the extent of the savings—measured in cases of disease, lives, and healthcare spending—is directly proportional to the rapid implementation of diagnosis and treatment practices such that achieving elimination targets by 2025 led to the overall best outcomes (Fig 1). This scenario requires an up-front investment in treatment and prevention, resulting in 1,400 lives saved and 1.1 billion USD in savings (Fig 2).

### Clinical analysis—Incident cases and cases averted

Accelerated elimination by 2025 resulted in the lowest number of cumulative incident cases of HCV (n = 11,600) over 2017–2030 (Table 3 and Fig 1); nearly 3.5 times as many cases occurred in the status quo scenario (n = 38,600), and 3,300–8,400 additional cases occurred in all other scenarios. Accelerated elimination averted approximately 20–30% more HCV incident cases (n = 27,000) compared to the one-year (n = 21,100) and two-year (n = 18,600) delayed elimination scenarios, and over 10% more cases averted compared to elimination (n = 23,700), with the status quo as the baseline. Total liver-related deaths (LRDs) were also the lowest under the accelerated elimination (n = 800) and elimination (n = 880) scenarios; however, all four elimination scenarios resulted in a substantial decrease in LRDs compared to the status quo, with 660–1,400 deaths averted (Table 3 and Fig 1). All elimination scenarios reduced the annual number of new LRDs to nearly zero during the study period, with the earliest occurring by 2023.

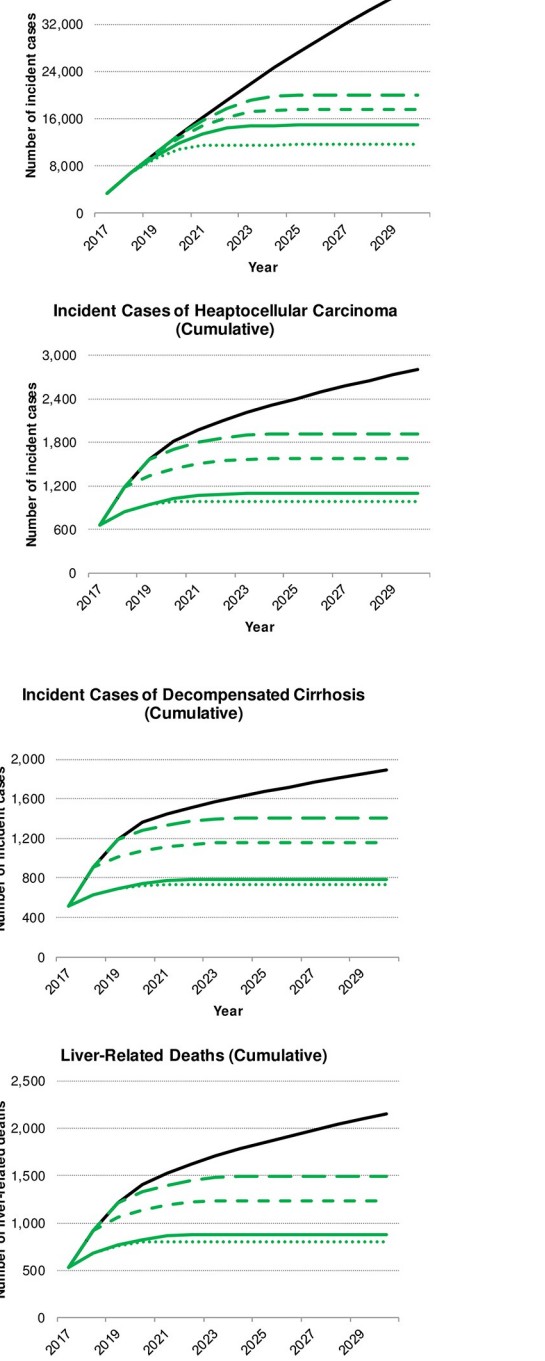

**Fig 1. Change in cumulative incident cases of HCV, hepatocellular carcinoma, decompensated cirrhosis and total liver-related deaths, by scenario, 2017–2030.**

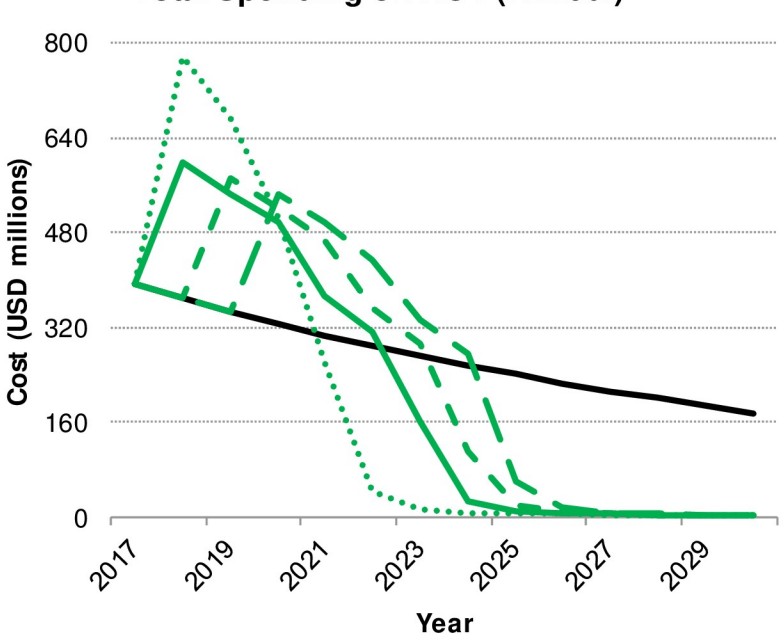

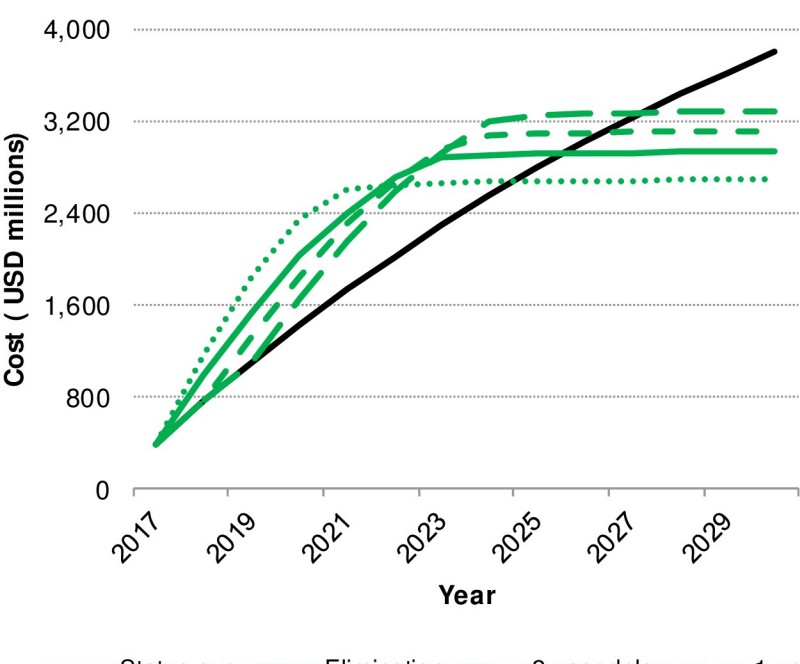

**Fig 2. Change in annual and cumulative HCV liver-related costs, by scenario, 2017–2030.**

The fewest cumulative cases of ESLD are expected to occur if elimination (or accelerated elimination) is initiated; 790 (n = 730) incident cases of DC and 1,100 (n = 990) incident cases of HCC are projected by 2030. With a one-year delay in implementation (or two-year delay), about 1,200 (n = 1,400) cases of DC and 1,600 (n = 1,900) cases of HCC are expected by 2030.

**Table 3. Cumulative HCV-related clinical and economic outcomes by scenario, 2017–2030.**

| Scenario | Status quo | Elimination | One-year delayed elimination | Two-year delayed elimination | Accelerated elimination by 2025 |
|---|---|---|---|---|---|
| **Clinical outcomes** | | | | | |
| **Incident cases of HCV (Cases averted***) | 38,600 (0) | 14,900 (23,700) | 17,500 (21,100) | 20,000 (18,600) | 11,600 (27,000) |
| **Total LRD (Deaths averted***) | 2,200 (0) | 880 (1,300) | 1,200 (920) | 1,500 (660) | 800 (1,400) |
| **New cases of DC** | 1,900 | 790 | 1,200 | 1,400 | 730 |
| **New cases of HCC** | 2,800 | 1,100 | 1,600 | 1,900 | 990 |
| **New cases of LT** | 100 | 20 | 30 | 40 | 17 |
| **Total cases of ESLD averted*** | 0 | (2,900) | (2,000) | (1,400) | (3,100) |
| **Economic outcomes (in USD millions)** | | | | | |
| **Total cost of liver-related complications** | 1,200 | 470 | 550 | 630 | 360 |
| **Total cost of extra-hepatic complications** | 1,800 | 630 | 760 | 890 | 450 |
| **Total treatment costs** | 750 | 1,700 | 1,700 | 1,600 | 1,700 |
| **Total screening costs** | 70 | 160 | 160 | 150 | 160 |
| **Total spending on HCV** | 3,800 | 2,900 | 3,100 | 3,300 | 2,700 |
| **Total costs saved compared to the status quo** | 0 | 860 | 680 | 510 | 1,100 |

Hepatitis C virus (HCV), decompensated cirrhosis (DC), hepatocellular carcinoma (HCC); liver transplant (LT), decompensated cirrhosis (DC), liver-related deaths (LRD), end-stage liver disease cases (ESLD) are a summation of DC, HCC and LT cases, United States dollar (USD).

*The number of cases averted describes cases for each outcome from the given scenario compared to the status quo (used as a baseline indicator).

As well, the elimination scenarios are projected to reduce cumulative new LTs between 60%–80% over 2017–2030, relative to the status quo. Depending on the swiftness of implementation, following the elimination plans could avert between 1,400–3,100 total cases of ESLD by 2030 (Table 3 and Fig 1). Notably, all elimination scenarios were estimated to reduce the annual incident cases of HCV, DC, HCC and LT to nearly zero by 2030. This is expected to be unattainable under the current standard of care.

## Cost analysis—HCV liver-related complications, treatment and screening

Over the 2017–2030 study period, all elimination scenarios resulted in cumulative medical cost savings associated with CHC infection in Korea: requiring between 2.7–3.3 billion USD in total HCV spending compared to a cost of 3.8 billion USD projected under current healthcare system. Accelerating elimination is expected to achieve the highest net savings of 1.1 billion USD, achieving reduced total spending compared to the status quo by 2025 (Table 3 and Fig 3). To achieve these cost savings, a significant investment in preventative measures must be implemented, with 160 million USD and 1.7 billion USD financing screening and treatment, respectively. In this scenario, 800 million USD would be spent on liver-related complications and EHM (360 and 450 million USD, respectively). Comparatively, continuing the status quo would result in spending only 70 million USD on screening and 750 million USD on treatment; however, liver-related complications and EHM would cost an additional 3.0 billion USD by 2030 (1.2 and 1.8 billion USD, respectively) (Table 3 and Fig 3). Implementing standard, one-year delayed, or two-year delayed elimination targets increased spending incrementally such that liver-related complications cost 470, 550 and 630 million USD, respectively, and EHM cost 630, 760, 890 million USD, respectively. Postponing elimination severely impacted net costs; a one-year delay dropped total cost savings over 40% and a two-year delay decreased net cost savings by half (Table 3).

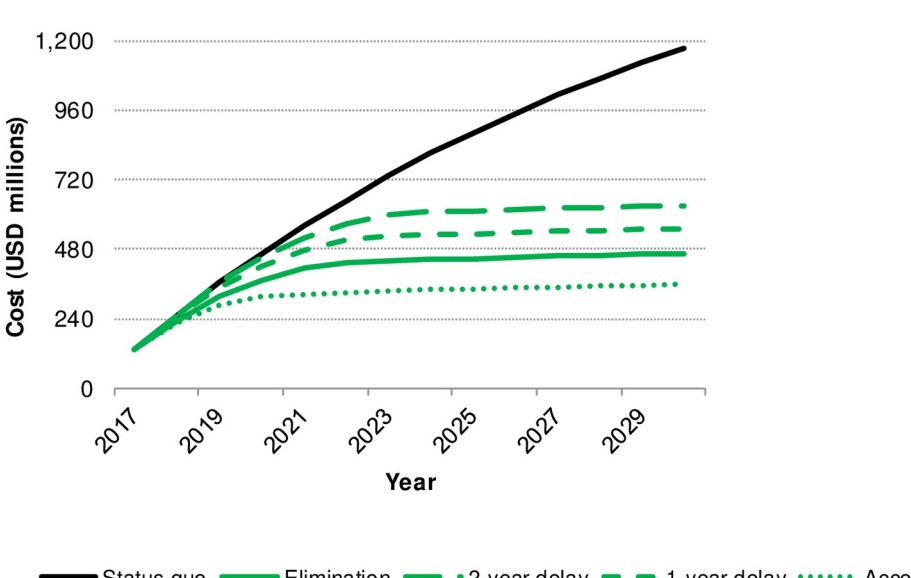

**Fig 3. Change in cumulative cost of HCV liver-related complications, by scenario, 2017–2030.**

## Discussion

Until about 2015, HCV was an unfamiliar disease in Korea, even for some physicians. However, two major turning points occurred that brought CHC to the forefront of the conversation. Firstly, a number of HCV-related healthcare-associated outbreaks reported between 2015 (70 infected patients) and 2016 (95 infected patients) increased disease recognition and prompted procedural changes [2, 15]. Beginning in September 2016, the Ministry of Welfare and Health began investigating all suspect cases of HCV, and in June 2017, HCV reporting became mandatory for all healthcare institutions. Prior to this, only 186 select medical institutions were required to report cases. The second turning point was the progress of treatment regimens with fewer side effects and markedly increased efficacy. Despite the availability of protease inhibitors beginning in early 2010, about 25–41% of patients could not achieve SVR, or ostensibly be "cured" [16]. In 2015, DAAs demonstrated SVR12 rates greater than 90% and were approved and reimbursed in Korean patients with genotype 1b. As such, Korean hepatologists began regarding CHC as a curable disease.

Increased national recognition of HCV indicates that 1) intervention efforts are needed now more than ever and 2) policymakers and healthcare workers may be particularly willing to discuss intervention efforts at this time. This analysis evaluates and compares different HCV elimination strategies with a goal of guiding next steps to reduce the burden of HCV in Korea.

The overarching difference between the four elimination scenarios and the status quo scenario relate to spending strategy over 2017–2030. In the status quo scenario, nearly 80% (3.0 billion USD) of the total HCV budget (3.8 billion USD) was spent on HCV liver-related complications. For all elimination scenarios, 55–70% (1.8–1.9 billion USD) of the HCV budget (2.7–3.3 billion USD) was instead invested in prevention. The difference between spending strategies is either an investment in prevention (elimination scenarios) or a focus on disease

mitigation and cure that is required due to lack of prevention, which also results in a less healthy Korea (status quo scenario). Postponing intervention by just one or two years led to worse clinical outcomes and a substantial increase in spending. The total number of LRD increased about 65% when comparing the standard elimination strategy to a two-year delay. Cost savings declined as the elimination plan was postponed, with savings decreased by half between the accelerated and two-year delayed strategies (Fig 2). The best-case scenario is implementation of accelerated elimination targets: saving money while reducing wasted expenditure on preventable complications. In this scenario, costing 2.7 billion USD, 70% of total funds (1.9 billion USD) would support national treatment and screening.

Regardless of the elimination scenario, additional spending is needed to continue or enhance HCV-related services to the public. Meeting accelerated elimination requires an increase in spending from the status quo by 98% in the first year of implementation. Comparatively, standard or delayed elimination warranted a 52–57% increase in spending in the first year of the program. While accelerated elimination requires a substantial initial investment, annual costs would reach that of the status quo in 2021 and continue to decline rapidly, well below the cost of the current strategy (Fig 2). In terms of surge capacity, there is a marked effect. In the third year of the accelerated elimination scenario (the year 2020), annual spending (500 million USD) would fall below the starting cost of the two-year delayed elimination plan (550 million USD) which begins in 2020. This demonstrates how rapid implementation is key, since postponing interventions by two years results in half the savings by 2030 and increased disease burden.

Historically, Korea has successfully leveraged government policy to manage infectious disease spread [17]. In 1995, Korea implemented universal hepatitis B virus (HBV) vaccination for newborns to prevent perinatal transmission. Coverage rates for the three-dose HBV series rapidly increased to 98% and have been sustained at this rate as of 2017 [18]. HBV vaccine introduction, along with other prevention methods, is attributed to decreasing HBV prevalence from 10% in the 1980s to about 2.9% in 2015 and diminishing the threat to middle-aged and younger populations [19, 20]. Although no vaccine exists for HCV, the success of strong government policy can be leveraged to target CHC in Korea. Case-finding is an integral first step, especially due to a diagnosis coverage of as low as 10% in the general population [21].

This study shows that accelerating elimination is the most effective method to reduce both the economic and disease burden of CHC. Similar result have been demonstrated in other analyses [22, 23]. Under the current standard of care, Korea is not projected to achieve HCV elimination by 2030. Current diagnostic and treatment practices must be increased nearly 4.5-fold and 3-fold, respectively, to achieve targets by 2025. In this respect, aggressive screening and linkage to care are the most important components of successful disease control, with the added benefits of saving money and lives. Unfortunately, no nationwide effort exists to diagnose asymptomatic cases, and HCV antibody testing is currently performed only at the patients' request. Implementation of a national screening program, however, would provide a broad platform to ramp-up and expand screening efforts. Local experts also agree that screening is the key step to reach elimination and suggest that including HCV antibody screening in the free annual health check-up program provided by KNHIS could be an important case-finding tool [12]. However, the government body is hesitant to adopt HCV testing into the national health check-up program because of potential costs. This study's results should serve as the backbone for policy and decision-making, demonstrating how aggressive prevention measures are designed to reduce future costs and increase the health of the public. Furthermore, KNHIS covers most currently available DAAs without restriction to cirrhosis or fibrosis level. Therefore, once patients are aware of their status,

they may be more likely to seek treatment due to the absence of a cost barrier. Given this, screening will be the most important element of successful disease elimination.

## Strengths and limitations

The Markov model is calibrated with Korea-specific HCV data using previously published disease progression rates, which strengthens the analysis. As a subcomponent, the Impact of Inaction platform is simplified for ease of use but also allows for the flexibility of manipulating multiple intervention scenarios. It is based in Microsoft Excel® to provide transparency and accessibility by external partners.

The analysis contains several limitations. Although prevalence estimates were obtained from the best available data in the literature, actual values may vary across settings and patient subgroups. Cost data were available for several additional medical conditions potentially associated to EHM; however, these were excluded due to a weak association with HCV and because they are not used in similar disease burden analyses. However, if these costs were included, we would expect even greater cost-savings under an elimination strategy. The societal outcomes of these scenarios, such as loss of productivity or quality-adjusted life years, were not considered. However, inclusion would likely support the case for aggressive prevention since decreased prevention is correlated with increases in morbidity and mortality. Additionally, due to the limits inherent in prediction, the model may not reflect observed results. Advancements in science could affect the types of diagnostics or treatment available for HCV in the upcoming decades, which could alter projected outcomes. As well, economic changes would influence the cost of HCV treatment and prevention in the future. Rather than operating as a detailed plan for intervention, results should support decision-making.

Factors influence the cascade of HCV care in Korea, such as linking diagnosed patients to care, treatment initiation and treatment retention, are poorly understood. Thus, even if accelerated screening was performed as designed in the scenarios, unexpected problems may occur regarding treatment uptake. Additional research may be needed to adequately understand the gaps in HCV care to inform and refine elimination strategies.

## Conclusions

By leveraging tools that quantify the impact of HCV screening and treatment interventions, Korea would avert a significant portion of incident cases, end-stage liver disease and death from HCV, and reduce costs. Postponing this intervention by just one or two years would lead to increased spending and fail to avert preventable hepatic complications and new infections, thus bringing considerable burden to patients and society.

## Supporting information

**S1 File.**
(SAS)

## Acknowledgments

The authors would like to thank Sarah Blach of CDA for providing medical writing and editing services in the development of this manuscript. AbbVie provided funding to CDA for this work.

## Author Contributions

**Conceptualization:** Yong Kyun Won, Kyung Sik Kang, Yuri Sanchez Gonzalez, Homie Razavi, Kwang-Hyub Han, Sang Hoon Ahn, Mi Young Jeon, Do Young Kim.

**Data curation:** Yong Kyun Won, Kyung Sik Kang, Yuri Sanchez Gonzalez, Mi Young Jeon, Do Young Kim.

**Formal analysis:** Kyung Sik Kang, Yuri Sanchez Gonzalez, Homie Razavi, Sang Hoon Ahn, Do Young Kim.

**Funding acquisition:** Yong Kyun Won, Do Young Kim.

**Investigation:** Yong Kyun Won, Kyung Sik Kang, Yuri Sanchez Gonzalez, Homie Razavi, Mi Young Jeon, Do Young Kim.

**Methodology:** Yong Kyun Won, Kyung Sik Kang, Yuri Sanchez Gonzalez, Homie Razavi, Ellen Dugan, Kwang-Hyub Han, Sang Hoon Ahn, Mi Young Jeon, Do Young Kim.

**Project administration:** Yong Kyun Won, Homie Razavi, Ellen Dugan, Mi Young Jeon, Do Young Kim.

**Resources:** Yong Kyun Won, Homie Razavi, Ellen Dugan, Sang Hoon Ahn, Mi Young Jeon, Do Young Kim.

**Software:** Yong Kyun Won, Homie Razavi, Ellen Dugan.

**Supervision:** Yong Kyun Won, Yuri Sanchez Gonzalez, Homie Razavi, Ellen Dugan, Kwang-Hyub Han, Sang Hoon Ahn, Do Young Kim.

**Validation:** Yong Kyun Won, Kyung Sik Kang, Yuri Sanchez Gonzalez, Homie Razavi, Ellen Dugan, Do Young Kim.

**Visualization:** Kyung Sik Kang, Homie Razavi, Ellen Dugan.

**Writing – original draft:** Homie Razavi, Ellen Dugan, Mi Young Jeon.

**Writing – review & editing:** Yong Kyun Won, Kyung Sik Kang, Homie Razavi, Ellen Dugan, Kwang-Hyub Han, Sang Hoon Ahn, Do Young Kim.

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
