## [Decision Letter · Decision Letter 0]

25 Nov 2019

PONE-D-19-29501

A tool to measure the impact of inaction toward elimination of hepatitis C: A case study in Korea

PLOS ONE

Dear Prof. Young Kim,

Thank you for submitting your manuscript to PLOS ONE. After careful consideration, we feel that it has merit but does not fully meet PLOS ONE’s publication criteria as it currently stands. Therefore, we invite you to submit a revised version of the manuscript that addresses the points raised during the review process.

We would appreciate receiving your revised manuscript by Jan 09 2020 11:59PM. To enhance the reproducibility of your results, we recommend that if applicable you deposit your laboratory protocols in protocols.io, where a protocol can be assigned its own identifier (DOI) such that it can be cited independently in the future. For instructions see: http://journals.plos.org/plosone/s/submission-guidelines#loc-laboratory-protocols

We look forward to receiving your revised manuscript.

Kind regards,

Tatsuo Kanda, M.D., Ph.D.

Academic Editor

PLOS ONE

Journal Requirements:

The design, study conduct, analysis, and financial support for the study were provided by AbbVie. AbbVie participated in the interpretation of data, review, and approval of the publication. All authors had access to all relevant data.

We note that you received funding from a commercial source: AbbVie.

Reviewers' comments:

Reviewer's Responses to Questions

**Comments to the Author**

1. Is the manuscript technically sound, and do the data support the conclusions?

Reviewer #1: Yes

Reviewer #2: Yes

2. Has the statistical analysis been performed appropriately and rigorously? 

Reviewer #1: Yes

Reviewer #2: Yes

3. Have the authors made all data underlying the findings in their manuscript fully available?

Reviewer #1: Yes

Reviewer #2: Yes

4. Is the manuscript presented in an intelligible fashion and written in standard English?

Reviewer #1: Yes

Reviewer #2: Yes

5. Review Comments to the Author

Reviewer #1: The authors investigated the clinical and economic impact of implementing national screening and treatment policies toward HCV elimination in Korea.

Conclusion of this study is really informative and useful for policy and decision-making.

Although study is well designed and manuscript is clearly organized, there are some points to be improved.

Reviewer #2: Comments to the Authors:

The authors, Yong Kyun Won. et al. reported the model compared the clinical and economic outcomes of current HCV-related interventions in Korea using the Markov model.

They concluded that elimination or accelerated elimination strategies would save 190 million USD or 1.2 billion USD by 2030, respectively, compared to the status quo, requiring an up-front investment in prevention that decreases spending on liver-related complications and death. The article is very interesting and can be useful in health and welfare.

Major comments:

1. The authors set five modelled scenario 1, status quo, 2, standard elimination, 3, delayed elimination by one year, 4, delaying elimination by two years, 5, accelerated elimination.

When using Marcov model, it is important to mention about Quality Adjusted Life Years.

Therefore, they should refer about the prognosis of Hepatocellular carcinoma and end stage of liver cirrhosis.

Fig1, liver-related death should be divided Hepatocellular carcinoma and end stage of liver cirrhosis independently.

2. They described a number of HCV-related healthcare-associated outbreaks reported between 2015 and 2016 increased disease recognition and prompted procedural changes.

They should add the number of estimated HCV antibody positive patients in Korea.

6. PLOS authors have the option to publish the peer review history of their article (what does this mean?). If published, this will include your full peer review and any attached files.

Reviewer #1: No

Reviewer #2: No

---

## [Author Response · Author response to Decision Letter 0]

22 Mar 2020

Feb, 2020 

Dear Reviewers of Plos One 

Thank you for your kind review for enhancing the quality of the manuscript. I and co-authors agree with your points and prepare the revised manuscript and revised figures for re-submission. 

From Major comments:

Comments: The authors set five modelled scenario 1, status quo, 2, standard elimination, 3, delayed elimination by one year, 4, delaying elimination by two years, 5, accelerated elimination.

When using Marcov model, it is important to mention about Quality Adjusted Life Years.

Therefore, they should refer about the prognosis of Hepatocellular carcinoma and end stage of liver cirrhosis.

We acknowledged this concern in the Discussions section, line 310. We mention that QALY was not addressed in this study, which is a limitation: “The societal outcomes of these scenarios, such as loss of productivity or quality-adjusted life years, were not considered.”

Fig1, liver-related death should be divided Hepatocellular carcinoma and end stage of liver cirrhosis independently.

We revise Fig1 as advised (Total, HCC, End stage liver Cirrhosis

Comments: They described a number of HCV-related healthcare-associated outbreaks reported between 2015 and 2016 increased disease recognition and prompted procedural changes.

They should add the number of estimated HCV antibody positive patients in Korea.

we add the number of patients in each outbreak in line 237-238. 

Discussion. This study showed that accelerating initiation of interventions to meet elimination targets by 2025 leads to the best profit. Please mention the current issue and how to resolve it

 For this comment, we added current issues and the way to resolve it in current circumstances of South Korea. Which is mainly located in second half of “discussion”.

Figures. It is difficult to see the difference because the lines overlap, so please change the scale (aspect ratio) to make it easier to understand. 

We re-manufactured the graphs more visible and clearer. Please see the new Figure files 

1) Please confirm future researchers can request access to the data underlying your study from the following contact information: (https://ocr.yuhs.ac/HPC/HPCIndex.aspx , Tel:+82-2-2228-0430, E-mail: irb@yuhs.ac). If not, please provide the appropriate contact information where future researchers can send data access queries.

Therefore, the author restricts the right for data access. DY Kim, corresponding author, has obtained approval for this research from his center’s institutional review board (https://ocr.yuhs.ac/HPC/HPCIndex.aspx , Tel:+82-2-2228-0430, E-mail: irb@yuhs.ac). Future researchers can request access to the data underlying your study from DY Kim, Corresponding author (DYK1025@yuhs.ac). 

Additional comment: However, please note PLOS ONE's data availability policy does not require authors to be the sole point of contact for fielding data access queries. Authors must provide a non author point of contact where data requests can be made. Therefore, please provide a non author point of contact where the data requests can be made (such as an institutional email for KNHIS).

The point of contact where the data requests was done and can be made further in KNHIS is as follow (Mr. HC Yoon, Data operation team #3, Dept. of Big data, Health insurance policy institute, KNHIS., Tel: +82-33-736-2473, (+82-33-811-2000 (English call center for foreigners)) Fax: +82-33-749-6337 E-mail: sapsary@nhis.or.kr )

2) Please also confirm the SAS code has been uploaded as a supporting information file. If it has not been already, please upload it as a supporting information file.

Data from KNHIS was also obtained under this IRB approval. The authors will also attach the SAS code used in the extraction of KNHIS data. The data file analyzed using the code is not provided because it is not open to the public by KNHIS’s regulation.

Hope this revision meeting your pointed issues. Once again, I and co-authors appreciate the kind review and please review our new revised version. 

Regards

DY Kim

Professor. Yonsei University, College of Medicine, Department of Internal Medicine, Seoul, Korea. 

E-mail: DYK1025@yuhs.ac

---

## [Decision Letter · Decision Letter 1]

9 Apr 2020

A tool to measure the impact of inaction toward elimination of hepatitis C: A case study in Korea

PONE-D-19-29501R1

Dear Prof. Do Young Kim,

We are pleased to inform you that your manuscript has been judged scientifically suitable for publication and will be formally accepted for publication once it complies with all outstanding technical requirements.

With kind regards,

Tatsuo Kanda, M.D., Ph.D.

Academic Editor

PLOS ONE

Additional Editor Comments (optional):

Reviewers' comments:

Reviewer's Responses to Questions

**Comments to the Author**

1. If the authors have adequately addressed your comments raised in a previous round of review and you feel that this manuscript is now acceptable for publication, you may indicate that here to bypass the “Comments to the Author” section, enter your conflict of interest statement in the “Confidential to Editor” section, and submit your "Accept" recommendation.

Reviewer #2: All comments have been addressed

2. Is the manuscript technically sound, and do the data support the conclusions?

Reviewer #2: Yes

3. Has the statistical analysis been performed appropriately and rigorously? 

Reviewer #2: Yes

4. Have the authors made all data underlying the findings in their manuscript fully available?

Reviewer #2: Yes

5. Is the manuscript presented in an intelligible fashion and written in standard English?

Reviewer #2: Yes

6. Review Comments to the Author

Reviewer #2: A whole report is well written. Their viewpoint that model compared the clinical and economic outcomes of current HCV-related interventions in Korea using the Markov models are scientifically sound.

7. PLOS authors have the option to publish the peer review history of their article (what does this mean?). If published, this will include your full peer review and any attached files.

Reviewer #2: No

---

## [Editor Report · Acceptance letter]

14 Apr 2020

PONE-D-19-29501R1 

A tool to measure the impact of inaction toward elimination of hepatitis C: A case study in Korea 

Dear Dr. Kim:

I am pleased to inform you that your manuscript has been deemed suitable for publication in PLOS ONE. Congratulations! Your manuscript is now with our production department. 

With kind regards,

on behalf of

Dr. Tatsuo Kanda 

Academic Editor

PLOS ONE